# Unveiling the Role of the Tumor Microenvironment in the Treatment of Follicular Lymphoma

**DOI:** 10.3390/cancers14092158

**Published:** 2022-04-26

**Authors:** Mariola Blanco, Ana Collazo-Lorduy, Natalia Yanguas-Casás, Virginia Calvo, Mariano Provencio

**Affiliations:** 1Medical Oncology Department, Hospital Universitario Puerta de Hierro-Majadahonda, 28222 Madrid, Spain; mariola.blanco@salud.madrid.org (M.B.); ana.collazo.externo@salud.madrid.org (A.C.-L.); virginia.calvo@salud.madrid.org (V.C.); 2Lymphoma Research Group, Medical Oncology Department, Hospital Universitario Puerta de Hierro-Majadahonda, IDIPHISA, CIBERFES, 28222 Madrid, Spain; nyanguas@idiphim.org

**Keywords:** tumor microenvironment, follicular lymphoma, therapeutic targets

## Abstract

**Simple Summary:**

Follicular lymphoma is the most common type of indolent non-Hodgkin lymphoma and is characterized by its heterogeneity and variable course. In addition to tumor cells, the immune microenvironment plays a fundamental role in the pathogenesis of the disease. Despite advances in treatment, responses vary among patients, and outcomes are often unpredictable: a subset of high-risk patients will be refractory to standard treatments or will develop a high-grade histology. In this review, we try to understand the crosstalk between follicular lymphoma B-cells and the tumor microenvironment as well as its impact on prognosis and the risk of transformation. We also highlight recent findings related to novel therapies developed to treat this complex disease, in which genetic mutations and microenvironment cells play a key role.

**Abstract:**

Follicular lymphomas (FL) are neoplasms that resemble normal germinal center (GC) B-cells. Normal GC and neoplastic follicles contain non-neoplastic cells such as T-cells, follicular dendritic cells, cancer associated fibroblasts, and macrophages, which define the tumor microenvironment (TME), which itself is an essential factor in tumor cell survival. The main characteristics of the TME in FL are an increased number of follicular regulatory T-cells (T_reg_) and follicular helper T-cells (T_fh_), M2-polarization of macrophages, and the development of a nodular network by stromal cells that creates a suitable niche for tumor growth. All of them play important roles in tumor angiogenesis, inhibition of apoptosis, and immune evasion, which are key factors in tumor progression and transformation risk. Based on these findings, novel therapies have been developed to target specific mutations present in the TME cells, restore immune suppression, and modulate TME.

## 1. Introduction

Follicular lymphoma (FL) is the most common indolent non-Hodgkin lymphoma (NHL) and accounts for 20% of all NHL cases [1]. The clinical presentation is heterogeneous, but in the majority of patients, the disease follows an indolent course over a long period, with repeated relapses leading to increasingly treatment-refractory disease. However, some patients have rapid disease progression and short survival, particularly those who transform to a more aggressive histology, typically diffuse large B-cell lymphoma (DLBCL), which has a transformation rate of around 2–3% patients per year in the U.S. [2,3]. Despite advances in treatment, most FL patients remain incurable. Management strategies include watch and wait, immunochemotherapy, and new targeted treatment options. The prognosis of FL remains heterogeneous, so prognostic indices are necessary to guide the decisions of physicians. As a result of an international cooperative study, the FL International Prognostic Index (FLIPI) was established in 2004, which included risk factors such as age, stage, tumor burden, serum lactate dehydrogenase (LDH), and hemoglobin. However, the FLIPI index was developed in the pre-rituximab era and was based on retrospective studies. For these reasons, and based on prospective data, the FLIPI2 index was created in 2009. This index also included bone marrow (BM) involvement and β2-microglobulin as prognostic factors [4]. FLIPI and FLIPI2 both have limitations, since they focus solely on clinical factors, leaving out biological factors such as genetic aberrations of tumor cells, the tumor microenvironment (TME), and the host response. Therefore, additional prognostic biomarkers that provide information about the underlying disease biology need to be developed.

Approximately 85% of FL cases harbor the t(14;18)(q32;q21) translocation in precursor B-cells that activate the expression of the anti-apoptotic B-cell lymphoma 2 protein (BCL2), although additional mutations are required for tumor onset [5,6,7]. Identifying the gene profile of FL has provided us with a more profound knowledge of the genetic alterations involved in pathogenesis, progression, risk of transformation, and treatment resistance. It has also enabled their integration into novel genetic risk models of prognosis such as M7-FLIPI, which considers the mutational status of seven genes (*EZH2, ARID1A, MEF2B, EP300, FOXO1, CREBBP, CARD11*) as well as clinical factors (FLIPI, ECOG scale), allowing high-risk patients to be identified [2].

FL develops in the GC of lymphoid follicles and resembles many of the morphological, immunophenotypic, and functional features of normal GC B-cells. However, the neoplastic follicles also contain non-neoplastic cells that constitute the TME such as CD4+ follicular helper T-cells (T_fh_), CD4+ follicular regulatory T-cells (T_reg_), CD8+ cytotoxic T-cells, follicular dendritic cells (FDCs), cancer-associated fibroblasts (CAFs), and tumor-associated macrophages (TAMs). All of them play a key role in the prognosis of FL [8].

Here, we review new advances in the knowledge of the FL microenvironment and their impact on prognosis, histologic transformation, and the development of new therapies.

## 2. Quiet Bystanders: TME Impact on FL Development and Prognosis

Understanding the main differences between normal and neoplastic lymph nodes (LNs) is essential. There is now growing evidence that crosstalk between lymphoma cells and TME cells is crucial for disease onset and progression.

In the non-neoplastic setting, B-cells must generate enormous variability in the antigen (Ag) recognition sites of their immunoglobulins (Igs), also known as B-cell receptor (BCR). For this purpose, they developed a mechanism that, in three independent steps (VDJ recombination, somatic hypermutation (SHM), and class switching) enables the production of Igs of almost unlimited specificity, although there is a risk of pathological mutations arising during this process. Naïve B-cells leave the BM after V(D)J recombination and migrate through the bloodstream to secondary LNs. B-cells that have encountered Ag enter the GC, which is formed by the expansion of selected clones within the FDC meshwork [9,10]. The proliferating GC B-cells are known as centroblasts, and they undergo SHM of the Ig variable region (IGV) genes, which alters the antigen affinity of the Ig that will be produced by the cell [11]. On the other hand, when centroblasts stop dividing, they become medium-sized cells called centrocytes, and if the IGV gene mutations have resulted in increased affinity for antigen, they present it to the T_fh_ cells via the major histocompatibility complex class II (MHCII). Centrocytes initially accumulate among the centroblasts and then migrate to the opposite pole of the GC. This causes the polarization of the GC into a “dark zone”, which contains centroblasts and closely packed centrocytes, and a “light zone”, which contains centrocytes, FDCs, and numerous T-cells. These two zones can be differentiated by the proliferation marker Ki67 as the “dark zone” contains the highly proliferating centroblasts and the “light zone” the low proliferating centrocytes. The selected centrocytes reenter the dark zone, where clonal expansion occurs and cells may cycle between the dark and light zones multiple times [12,13]. Finally, centrocytes will become either plasma B-cells or memory B-cells. The interaction with CD23, which is expressed by FDCs, seems to be important in the differentiation into plasma cells, stimulating class switching. On the other hand, interaction with T_fh_ cells through the CD40–CD40 ligand appears to be important for generating memory B-cells [14,15].

This GC reaction is highly dependent on the TME. T-cells are predominantly located in the light zone and most are T_fh_ cells, which are important for selecting B-cells for entry and proliferation within the GC. T_reg_ cells are present in low numbers; they exert negative regulatory effects on both B- and T-cells and are needed to stop the GC reaction, preventing excessive immune responses [16]. FDCs are large cells with delicate nuclear membranes that derive from mesenchymal origin. They have surface complement receptors (CD21) and Fc receptors (CD23) that bind free antigen and antigen–antibody complexes for presentation to B-cells. In normal lymph nodes, the role of FDCs and CAFs is to build a network to support the GC reaction. In contrast, macrophages (CD68+) are typically phagocytic, containing apoptotic debris from B-cells that have failed to express a surface Ig molecule [17]. Macrophages can be polarized into M1 (inflammatory phenotype) or M2 (anti-inflammatory), resulting in distinct cytokine production or T-cell function (Th1 and Th2).

Many of the components of the FL microenvironment mimic those present in normal GCs, but some important differences may contribute to tumor-cell survival. The role of TME in FL can be defined in two directions: supporting tumor growth and suppressing the antitumor immune response [18]. We summarize the main characteristics of B-cells and TME cells in FL below. Table 1 recapitulates the main cells involved in the TME of FL.

### 2.1. B-Cells

The histopathology of FL is closely related to key events in normal B-cell development and differentiation. As previously mentioned, in approximately 85% of FL cases, the malignant B-cells harbor a translocation t(14;18)(q32;q21) that occurs as a consequence of aberrant VDJ recombination in a progenitor B-cell in the BM, in contrast to normal GCs [19,20]. By juxtaposing BCL2 next to the immunoglobulin-heavy chain enhancer, this translocation leads to a constant expression of BCL2, which is observed in almost all FL (≥25% of FL cells) but not in normal GCs, resulting in the inhibition of apoptosis and favoring the selection of these clones in the GCs of secondary lymphoid organs [21]. The t(14;18) translocation in B-cells increases the risk of developing FL, although it is not sufficient and the acquisition of additional mutations is necessary for neoplastic transformation and disease progression, since the t(14;18) translocation has also been observed in GC B-cells of healthy individuals [22]. FL B-cells undergo repeated re-entry into GCs, so this leads to the acquisition of these additional aberrations that enable FL to develop.

Unlike normal GCs, in FL, there is no differentiation into a “dark zone” and a “light zone”, but there is a predominance of centrocytes interspersed with some centroblasts [23]. FL B-cells express many of the antigens that are found on normal GC B-cells and that are associated with interactions with T-cells and dendritic cells such as costimulatory molecules CD80, CD86, CD40/43/44, and CD70 [24]. Furthermore, they express surface Igs that have undergone SHM of the IGV genes and, in approximately 20–50% of cases, have undergone Ig class switching. In a high proportion of FL cases during SHM, new mutations can be acquired. These mutations introduce consensus sequences of unusual mannosylated glycans, which are not seen in normal B-cells and can bind to lectins on FDCs and macrophages, enabling them to survive in the GC environment in the absence of cognate antigen [25,26].

### 2.2. T-Cells

In the neoplastic follicles, T-cells are less numerous than in reactive follicles and they are randomly distributed, in contrast to the concentration in the light zone that characterizes normal lymph nodes [27]. CD4+ T-cells (T_reg_ and T_fh_) are predominant with a higher CD4:CD8 ratio, T_reg_ cells being more numerous than T_fh_ cells. Both types of cells are fundamental to providing tumor support and facilitating immune evasion.

FL-associated T_fh_ lymphocytes are defined by the expression of the chemokine receptor CXCR5, the inducible T-cell co-stimulator (ICOS), the programmed cell death 1 (PD-1), and the transcription repressor BCL6. These cells produce an excess of chemokines (IL-4, IL-17, IL-21, and IFN-γ) and overexpression of CD40L, supporting B-cell viability, inhibiting apoptosis, and influencing the TME. Moreover, T_fh_ cells expressing IL-4 and CD40L can also induce FL cells to produce CCL17 and CCL22, thereby promoting the active migration of T_regs_ [15,27].

On the other hand, T_reg_ cells are characterized by the expression of the transcription factor FOXP3 and they have a negative regulatory effect on B- and T-cells, playing a pro-tumor role due to their immunosuppressive activity, which hampers CD8+ T-cell activation [28]. Interestingly, FL patients are particularly rich in CD4+ T-cells that harbor CXCR5^high^ ICOS^high^ PD1^high^ and FOXP3+/CD4+/CD25+ phenotypes, which are called follicular regulatory T-cells (Tfr). T_fr_ cells have greater suppressive capacity than T_regs_ in normal LNs because they upregulate cytotoxic T-lymphocyte antigen 4 (CTLA-4), IL-10, and the glucocorticoid-induced tumor necrosis factor receptor-related protein (GITR). This pattern has been described as being a predictor of shortened overall survival (OS) [29].

### 2.3. Stromal Cells, Follicular Dendritic Cells and Macrophages

Stromal cells and macrophages are key factors in the pathogenesis of FL, but their association with prognosis remains controversial.

FDCs and CAFs are stromal cells that phenotypically and functionally differ from their normal counterparts, presenting a niche-based model of oncogenesis attributed to the dynamic coevolution of cancer and stromal cells. FDCs are particular Ag-presenting cells (APCs), and they present intact Ag–Ab complexes on their cell surface, which enable survival and induce the differentiation of FL cells. They can also trigger T_fh_ recruitment by the production of CXCL13, which interacts with CXCR5. The FDCs of almost all FL express CD21, while only a small subgroup express CD23, which can be expressed by FDCs and B-cells. In contrast, CAFs secrete components of the extracellular matrix such as laminin, fibronectin, and collagen. The nodular network built in the neoplastic setting infiltrates follicles, creating a suitable niche even in extranodal sites and BM, suggesting that they are recruited by the neoplastic cells [15]. The crosstalk between FDCs, CAFs, and FL B-cells is crucial to promote tumor cell survival. They are responsible for B- and T-cell migration thanks to the upregulation of CXCL12, CXCL13, and BAFF signaling as well as adhesion through interactions between adhesive molecules such as VCAM-1 and VLA-4. Through these interactions, stromal cells protect malignant cells against apoptosis induced by chemotherapy or rituximab. They can also induce malignant monocyte recruitment and their differentiation to M2-protumoral macrophages through the secretion of CCL2 and CSF-1 [30,31].

TAMs are typically non-phagocytic, with a low proliferation fraction, and they do not undergo apoptosis. They express CD163+, the Fc fragment of IgG, and C-type lectin domains. These cells can display two different phenotypes: M1- and M2-polarization. This polarization is controlled by tumor cells within the TME and is dynamic throughout cancer development. M1-macrophages are typically tumor-suppressing cells that act in the TME by recruiting CD8+ T-cells and NK cells. These CD8+ and NK cells express high levels of cytokines and chemokines, recruiting other immune cells and favoring the signaling of anti-tumorigenic pathways [32]. On the other hand, the M2-phenotype is associated with tissue remodeling, angiogenesis, and progression, making them an attractive target for therapies [33]. These M2-macrophages secrete immunosuppressive molecules into the TME, suppress T-cell mediated anti-tumor responses and recruit T_reg_ lymphocytes, facilitating tumor proliferation and immune evasion [32,34]. TAMs also play a key role in tumor angiogenesis by secreting VEGFA, which can stimulate the chemotaxis of endothelial cells and macrophages [35]. In addition, they promote the epithelial–mesenchymal transition process, which enables cancer cells to leave the tissue site, favoring the initiation of metastasis [36]. However, TAM’s polarization state is not fixed. At early stages of tumor development, M1-macrophages are predominant, while the M2-phenotype is more frequent in the advanced setting where tumor proliferation and invasion increases [32]. For the time being, the role of TAMs for the prognosis of FL remains unclear. In the pre-rituximab era, some studies have suggested that macrophage infiltration was associated with lower survival, probably due to their M2-polarization. Nevertheless, the addition of rituximab to chemotherapy modified their prognostic impact, and, although the mechanism is still not well-known, it is likely to be associated with antibody-dependent cell-mediated toxicity (ADCC) [37,38]. The antitumoral activity of rituximab is dependent on interactions with the effector cells that have Fc receptors (neutrophils, natural killer cells, and macrophages). Because tissue macrophages are critical for B-cell depletion after anti-CD20 antibody therapy, it is plausible that there is a relationship between TAM content and the efficacy of rituximab [39]. The Finnish and French groups showed that the addition of rituximab to chemotherapy reversed the negative prognostic impact of high macrophage content to favorable, reporting a survival benefit in the rituximab arm [40].

### 2.4. Neutrophils

Little is known about the interactions between FL cells and neutrophils although they are important players in the innate immune system. Unlike other hematological cells, they are predominantly present in peripheral blood. In preclinical studies, tumor associated neutrophils (TANs) have demonstrated a reduced cytotoxic effect of chemotherapy through the interaction of CD11b/ICAM-1 with CD44 of malignant B-cells. An increased number of TANs in FL patients has been correlated with poor prognosis and weak treatment responses. These findings suggest that TANs could become a new targeted therapy [41].

In summary, FL B-cells are surrounded by a network of supportive cells that take part in the development of tumor cells, treatment resistance, prognosis, and histological transformation.

## 3. FL-Morphing: TME and Risk of Transformation

The outcome for FL patients dramatically improved with the incorporation of chemo-immunotherapy. However, despite the indolent course of FL, a subset of patients may evolve with rapid disease progression, transforming into high-grade lymphomas. Genetic and individual tumor factors have been associated with the risk of transformation. Regarding individual tumor factors, the intrinsic characteristics of tumor cells, the composition of the TME and patient characteristics are crucial aspects that must be taken into account.

Gene expression profiling studies have revealed many genetic alterations that are upregulated in this subset of patients with aggressive diseases and that may be involved in the risk of transformation. These include changes in genes regulating proliferation, control of cell cycle, DNA damage response, and apoptosis such as *MYC*, the BCR and TLR pathways, *p53*, *FOXO1,* and *BCL6*. *MYC*-related genes play an important role in tumor transformation, but not all transformed cases have an increased *MYC* signature [3]. Genetic alterations in TME cells have also been described. Expression of *PMCH*, *ETV1*, and *NAPMT* on FL-infiltrating T-cells leads to a decrease in T-cell motility and appears to be a key factor in FL transformation in preclinical studies [42,43,44].

Focusing on TME, PD1+ T-cells and FDCs have been identified as independent predictors of transformation. The pattern of distribution of these cells seems to be more important than the number itself. The FDC network localized in the follicle was correlated with a shorter time to tumor transformation (TTT), progression free survival (PFS), and OS. Those PD1+ T-cells with a diffuse pattern of infiltration had a worse outcome than those with a follicular pattern, and were more likely to undergo early transformation [15,45]. A high PD-1+ cell content was predictive of favorable outcome of FL patients, whereas a marked reduction was observed in transformation [46]. In addition, a decrease in the number and follicular distribution of FOXP3+ T_reg_ cells also seems to be an important factor [28]. In relation to CAF, pyruvate secretion seems to support the survival of primary FL cells and may be associated with a higher risk of transformation. Pyruvate is the fundamental metabolite secreted by CAF, resulting in increased use of the citric acid cycle in lymphoma cells, leading to decreased redox regulation. A preclinical study performed in lymph node samples of malignant lymphoma patients proved that the addition of pyruvate to culture media, with a median concentration of 0.25 mmol/l, contributed significantly to the survival of primary FL cells [47]. Further research to uncover the role of CAFs in the lymphoma microenvironment is needed.

## 4. Hunting the Achilles’ Heel: Therapeutic Strategies Targeting the TME

Although many asymptomatic patients with indolent disease may not require early treatment, most of them will require therapy during the course of their disease. Nowadays, having gained a deeper knowledge, the TME itself has become a therapeutic target and is considered as such in existing and upcoming therapeutic approaches (e.g., PD-1/PD-L1 inhibition, lenalidomide, and bispecific antibodies, targeting both the tumor cells and cells from the TME). Figure 1 illustrates potential therapeutic targets in FL. Table 2 describes current therapies against the TME in FL.

### 4.1. Immune Checkpoint Blockade Therapy in Follicular Lymphoma and the Role of PD-1/PD-L1 Expression

Advances in the characterization of immune landscapes in solid tumors have led to them being classified as those exhibiting, or not, a T-cell inflamed phenotype. T-cell inflamed tumors are morphologically characterized by immune-cell infiltration and are highly sensitive to checkpoint-blockade therapy, whereas T-cell non-inflamed tumors are typically resistant. Nevertheless, it has not been easy to extrapolate this concept to lymphomas because most of them originate in the secondary lymphoid organs in which the immune cells reside. Genomic studies and observations from clinical trials have enabled us to classify lymphomas with respect to these two subgroups. Classic Hodgkin lymphoma and primary mediastinal large B-cell lymphoma are typically part of the inflamed subtype, whereas DLBCL, FL, chronic lymphocytic leukemia, and Burkitt lymphoma present with non-inflamed environments. More studies on the FL microenvironment are needed to determine whether there is an additional subset of inflamed phenotypes that might benefit from checkpoint inhibitors [3,48]. The main goal of checkpoint-blockade therapy is to stimulate the immune response while simultaneously inhibiting immune suppression. There are immune-checkpoint activators (CD40L, OX40, CD27, CD28, and 4-1BB/CD137) and inhibitors (CTLA-4, PD-1, LAG-3, TIM-3, and TIGIT), and both regulate T-cell activity. Here, we review the main immune-checkpoint inhibitors (ICIs) that may be relevant in FL.

The introduction of PD-1/PD-L1 ICIs, with their unprecedented clinical efficacy, has dramatically changed our appreciation of the natural history of many solid tumors [8]. Previous clinical studies showed that PD-L1 expression was associated with the clinical response to PD-1 blockade and prognosis in solid tumors [49,50]. In lymphomas, these therapies have been approved and are currently widely used to treat refractory classic Hodgkin lymphoma [51]. The expression of PD-L1/PD-1 in tumor cells and tumor-infiltrating T-lymphocytes (TILs) varies between multiple subtypes of lymphoma including FL. PD-1^high^ T-cells are characterized by their ability to support tumor growth without TIM-3 expression, while PD-1^low^ T-cells have an exhausted phenotype and usually express TIM-3. This exhausted phenotype is characterized by reduced cytokine production and cell-signal transduction. Nevertheless, PD-1 expression is not sufficient to distinguish exhausted from activated T-cells, and immune escape in FL involves many other pathways in addition to PD-1 [52]. In terms of clinical development, the PD-1 axis has been the most thoroughly explored. Pidilizumab is an anti-PD-1 antibody that has shown promising activity in relapsed/refractory (R/R) FL in combination with rituximab in a phase II trial of 32 patients, showing an overall response rate (ORR) of 66% and a complete response rate (CR) of 52% [53]. Based on the promising activity of nivolumab in the phase I study CheckMate 039, where an ORR of 40% was observed in R/R FL, the phase II trial CheckMate 140 assessed nivolumab in 92 patients with R/R FL. Unfortunately, nivolumab showed very limited activity as monotherapy (ORR 4%) [54]. Further clinical trials combining anti-PD-1 with anti-CD20 antibodies or targeted therapies are in progress in FL patients, but at this point, the role of the PD-1 inhibitors in the therapeutic arena of FL remains unclear.

CTLA-4 is a homolog of CD28 and binds to CD80/CD86. It is usually expressed in T_reg_ and activated T-cells, and is a negative regulator of T-cell function that dampens antitumor immune responses. Using a monoclonal antibody directed against CTLA-4 improves immune function. In a phase I trial of ipilimumab in combination with rituximab in patients with R/R B-cell NHL (with 39% of FL included), a manageable toxicity and encouraging efficacy were observed (ORR 58%). The ratio of CD45RA T_regs_ was significantly higher in responder patients [55].

Additional ICIs have been explored in FL. Lymphocyte activation gene 3 (LAG3/CD223) is upregulated on activated TILs, which induces the inhibition of T-cell activity by reducing their ability to produce cytokines and granules, and eventually, its anergy. LAG-3 expression on intratumoral T-cells was correlated with a poor outcome in FL patients [56]. On the other hand, the cell immunoglobulin and ITIM domain (TIGIT) targets the immune response at different levels through its action on antigen-presenter cells, CD8+ T-cells, and T_reg_ cells. As mentioned before, anti-PD-1/PD-L1 therapy alone is probably not the best approach to achieve strong responses in FL patients. Thus, therapeutic strategies with anti-LAG-3/anti-TIGIT, alone or in combination with anti-PD-1 antibodies, are under investigation in order to improve clinical outcomes in FL patients. In addition, emerging therapies targeting immune checkpoint activators such as anti-CD40 and anti-CD137 are also being explored in clinical trials in combination with other ICIs and anti-CD20 therapy [57]. Urelumab, an anti-CD137 antibody, has been tested in a phase Ib study, either alone or in combination with rituximab, in patients with R/R B-cell lymphomas including FL (17 patients). The ORR was 12% with urelumab alone, and 35% with the combination in the subset of FL patients. The responses were durable and had a manageable toxicity profile. However, the clinical activity was not enhanced with the combination treatment relative to rituximab alone [58].

The poor responses obtained with ICIs in FL could be explained by the heterogeneity of PD-1+ expression on T-cells. Drug combinations seem to achieve higher response rates in comparison to ICI monotherapy in FL. As above-mentioned, several clinical trials exploring new treatment strategies are currently ongoing, and preliminary results are encouraging.

### 4.2. Immunomodulatory Agents

Lenalidomide is an oral immunomodulatory drug with effects on the immune and nonimmune microenvironments including angiogenesis and direct antitumor activity with the inhibition of tumor cell proliferation. This drug targets the ubiquitously expressed E3 ubiquitin-ligase cereblon, inducing ubiquitination and the degradation of many targets. It also stimulates T-cell- and NK-cell-mediated cytotoxicity in B-cell lymphomas including FL [33]. In T-cells, administering lenalidomide results in enhanced costimulation signaling (upregulation of CD80/CD86, CD1c, CD40, and downregulation of PD-L1) and increased IL-2 secretion, so that tumor cells will be more visible to the surrounding immune effector cells. It has also shown the potential to enhance the rituximab effect via the NK-cell-mediated and monocyte-mediated antibody-dependent cytotoxicity (ADCC/ADCP) mechanism, which stimulates dendritic cells and alters the cytokine microenvironment. Its combination with rituximab has shown promising results in clinical trials, whereby the response rates were high and durable (ORR 74–80%) in the R/R setting [59,60]. For this reason, this combination has recently been approved for R/R indolent B-cell lymphoma including FL. Combinations with obinutuzumab, a newly developed anti-CD20 monoclonal antibody, have also been investigated in a phase II study, which achieved similar response rates to that with rituximab, so it may be another option in the near future [61,62]. In summary, immunomodulatory agents will become important building blocks of combinatorial strategies with monoclonal antibodies, novel targeted immune-checkpoint inhibitors (anti-PD-1/PD-L1, anti-CD47), and emerging cellular immunotherapies for forming long-lasting immune synapses.

### 4.3. Targeted Therapies

Targeted therapies against TME have the theoretical potential to reverse the tumor-supportive function of the FL microenvironment as well as improve the function of immune effectors and to disrupt angiogenesis. Here, we review the main targeted therapies developed in FL and their relationship with the TME.

FL cells behave in a manner similar to normal B-cells, so they remain dependent on BCR signaling. Thus, the use of BCR inhibitors such as PI3K inhibitors is an attractive approach. The PI3K pathway is downstream from the BCR and is vital for the survival of FL. There are four class I PI3K isoforms in mammals: p110α and p110β are characterized by ubiquitous tissue distribution, while p110γ and p110δ are mainly expressed in hematopoietic cells. To date, four PI3K inhibitors have been approved for R/R FL: idelalisib, duvelisib, copanlisib, and umbralisib. The oral PI3Kδ inhibitor idelalisib was the first PI3K inhibitor to be approved. It diminishes tumor cell proliferation and reshapes the immune microenvironment, reducing FDC-induced genes related to angiogenesis, extracellular matrix formation, and transendothelial migration in sensitive patients. This compound interferes with the B-T immunological synapses via the CD40/CD40L axis and affects T_reg_ and T_fh_ recruitment through CCL22 downregulation [63]. Idelalisib was evaluated in a phase II trial in patients with indolent R/R NHL, in which it showed rapid and durable responses, with an ORR of 57%. However, adverse events, mainly diarrhea, elevation of aminotransferase levels, and pneumonitis, were frequent [64]. Duvelisib is an oral dual PI3Kγδ inhibitor that targets both tumor and microenvironment cells as PI3Kγ is expressed by CD4+ T-cells and M2 macrophages. This compound repolarizes the immunosuppressive M2-like phenotype to the inflammatory M1-like phenotype in macrophages, inhibiting tumor-cell proliferation and survival. The ORR achieved with this drug was 42% (lower than that with other PI3K inhibitors) [65,66]. Copanlisib, another PI3K inhibitor, is an intravenous pan-class I PI3K inhibitor that predominantly acts against PI3K α/δ, an agent with inhibitory activity that induces apoptosis and inhibition of the proliferation of primary malignant B-cell lines. The ORR obtained with copanlisib was 59% and the safety profile was more favorable than those of idelalisib and duvelisib, with lower rates of severe transaminitis, colitis, and colonic perforation [67]. Umbralisib is an oral dual inhibitor of PI3Kδ and casein kinase-1ε (CK1ε). The ORR achieved with this drug in the phase II UNITY study was 45%, with a favorable benefit–risk profile. It has been approved in the fourth-line setting in R/R FL patients, as the median number of prior systemic therapies in the UNITY trial was three [68]. The newer-generation PI3K inhibitors, copanlisib and umbralisib, appear to be better tolerated, although larger-scale studies and real-world data are required to fully understand their tolerability and position among the other two PI3K inhibitors [63]. In the context of therapies targeted against TAMs, the colony-stimulating factor 1 receptor (CSF-1R) is a new target for inhibiting FL-TAMs crosstalk. CSF-1 regulates the survival, proliferation, and chemotaxis of macrophages, and supports their activation. It is relevant in high-risk patients because of its association with aggressive disease and the risk of transformation [33]. Pexidartinib inhibits CSF-1R, diminishing the viability of M2-macrophages and repolarizing them to M1-macrophages, thereby giving rise to an antitumor phenotype. The combination of pexidartinib and rituximab is currently under investigation in preclinical studies [69]. Another therapeutic approach is the administration of an antibody drug that blocks CD47, a receptor expressed in tumor cells that prevents phagocytosis and forms a complex with signal-regulatory protein α (SIRP-α), which is expressed in macrophages. Anti-CD47 antibodies, since they are promising drug candidates, are now undergoing clinical trials, especially in combined immunotherapeutic regimens that include rituximab [70]. Hu5F9-G4 is a humanized, IgG4 isotype, CD47-blocking monoclonal antibody. In a phase Ib study, the combination of Hu5F9-G4 and rituximab elicited responses in half of the patients with R/R aggressive and indolent lymphomas, and has an acceptable safety profile [71].

### 4.4. Epigenetic Regulators

As previously mentioned, genetic alterations are necessary for the development and transformation of FL. In addition to aberrations in multiple signaling pathways, deregulation of epigenetic components ranging from histone modifications to noncoding RNAs appear to have a pivotal role. The most common aberrations in the epigenetic regulators are *KMT2D/MLL2*, *CREBBP*, *EP300*, *EZH2*, and *HIST1H1*; together, they account for 95% of cases. These mutations modify histones and are thereby involved in epigenetic regulation by influencing key pathways such as BCR, PI3K/AKT, Toll-like receptor (TLR), mTOR, JAK/STAT, MAPK, CD40/CD40L, and chemokine/interleukin signaling. In addition, mutations can involve genes such as *MYC*, *BCL6*, *TP53*, *FOXO1*, *MYD88*, *STAT3*, *MDM2,* and *CDKN2A/B*, which control cell survival [3]. Research into these genomic aberrations through gene expression profiling and their relationship with TME has enabled us to develop new targeted therapies.

*KMT2D* encodes a histone methyltransferase of H3K4 and is a tumor suppressor. Its functional impairment causes an expansion of GC B-cells and promotes lymphomagenesis [72]. However, no impact on the TME has so far been described. *CREBBP* and *EP300* are histone acetyltransferases, and their mutations inactivate genes that are direct targets of the BCL6–HDAC3 complex, leading to a downregulation of MHCII and impaired p53 activation, blocking exit from the GC. HDAC3 inhibition partially restores the immune responses and may represent a new therapeutic approach in FL [73]. Some clinical trials have explored treatment with pan-HDAC in monotherapy or combined with other drugs (mainly immune-checkpoint blockade). Vorinostat and abexinostat in monotherapy have respectively shown ORRs of 47% and 63% in phase II trials in the R/R setting [74,75,76]. In contrast, *EZH2* functions as a histone methyltransferase that methylates H3K2. *EHZ2* mutations are involved in the maintenance of the GC reaction as well as in TME interactions, potentiating their dependence on FDCs and playing a role in developing natural killer (NK) cells. *EZH2* mutations are associated with a reduced risk of early relapse after chemo-immunotherapy. The U.S. Food and Drug Administration (FDA) has recently approved the use of the selective *EZH2* oral inhibitor tazemetostat for adult relapse/refractory (R/R) patients with *EZH2*-mutated tumors and patients with R/R FL wild-type EZH2 for whom there are no satisfactory alternative treatment options. Tazemetostat showed an ORR of 63–71% and a PFS benefit compared with the non-mutated cohort of patients (13.8 vs. 11.0 months) [33,77]. Combinations of tazemetostat with other therapeutics such as rituximab and lenalidomide are currently under investigation and it is hoped that they will improve the survival outcomes relative to the single agent tazemetostat [78]. Mutations of *TNFRSF14* (tumor necrosis factor receptor superfamily 14), also known as herpesvirus entry mediator A (*HVEM*), are another common event in FL, occurring in approximately 50% of cases. *HVEM* limits T-cell activation via the ligand of BTLA (B- and T-lymphocyte attenuator) and induces a tumor-supportive microenvironment characterized by exacerbated lymphoid stroma activation. *HVEM* is in the minimal common region of the chromosome 1p36 deletion, which is associated with worse prognosis in FL. A preclinical CAR-T construct has been developed to continuously produce soluble TNFRSF14 in the microenvironment, which restores its inhibitory function [79].

### 4.5. New Horizons in the Treatment of Follicular Lymphoma

#### 4.5.1. Bispecific Antibodies

Another attractive strategy is the development of T-cell-bispecific antibodies. Using protein engineering techniques, researchers have developed bispecific antibodies that can bind two target antigens rather than one. By targeting one antigen on malignant cells (CD20) and another on T-cells, bispecific antibodies can act as a bridge to bring immune effector cells in close proximity to malignant cells, which results in cell-mediated cytotoxicity. Currently, the CD3/CD20 bispecific antibodies mosunetuzumab (RG7828), glofitamab (RG6026), and odronextamab (REGN1979) are undergoing phase I clinical trials and preliminary results have shown an impressive efficacy in heavily pretreated FL patients [76,77,78]. A recent update of the phase I/Ib trial that evaluated mosunetuzumab has shown favorable and durable responses rates, with an ORR of 64% and a CR of 42% in patients with indolent NHL including FL. It has also shown responses after failure of CAR T-cell therapy, with few patients (22%) achieving CR [30]. Overall, the bispecific antibodies show promise with respect to their activity in heavily pretreated patients with FL, in whom they have greater availability than CAR-T.

#### 4.5.2. CAR-T-Cell Therapy

Adoptive transfer of chimeric antigen receptor (CAR)-T-cells targeting antigens on lymphoid cells has revolutionized the treatment of advanced B-cell malignancies. Anti- CD19 CAR-T-cell therapies are known to have remarkable efficacy in B-cell acute lymphoblastic leukemia, chronic lymphocytic leukemia, and refractory DLBCL. Based on these results, CAR-T-cells have been explored in largely incurable lymphomas including FL. More recently, the FDA approved the use of axicabtagene ciloleucel (axi-cel) based on results from the ZUMA-5 phase II study for adult patients with R/R FL after two or more lines of systemic therapy. The ORR of axi-cel was 94%, with 80% of patients achieving CR, but with an increased risk of secondary toxicities, as 85% of patients had a ≥grade 3 adverse event including neurotoxicity (15%) [79]. Based on the activity of the CAR-T with its durable clinical benefit, this approach has the potential to change the natural history of FL, although new interventions with agents to mitigate the toxicity are needed. However, CAR-T-cell therapy can become ineffective due to CD19-antigen loss or downregulation and the suppressive impact of the TME, reducing their efficacy. This is the reason why further efforts are focused on targeting the lymphoma cells and T_fh_ cells within the nodal stroma. Anti-CXCR5 CAR-T-cells offer a promising treatment strategy for nodal B-cell NHLs through the simultaneous elimination of lymphoma B-cells and T_fh_ cells of the tumor-supporting TME [80,81].

## 5. Conclusions

Understanding the differences between normal and neoplastic follicles including those in the TME has provided new insights into lymphoma development and progression. While BCL2 overexpression remains the hallmark of FL carcinogenesis, additional aberrations such as epigenetic mutations are essential in this context. Notably, the increasing amount of research in the area has highlighted the role of the TME in the pathogenesis of FL, reinforcing the need to develop targeted therapies or novel drug combinations against the TME. These new therapies have shown encouraging results and have expanded treatment options, improving outcomes in FL patients. This is a significant challenge, as in most patients with FL, the disease follows an indolent course and studies take many years to come to fruition. 

## 6. Future Directions

This review focuses on the network of associations between TME and FL neoplastic cells as well as their involvement in prognosis, the risk of transformation, and advances in treatment. New therapies targeting tumor and non-tumor cells are currently being developed. These make use of novel monoclonal antibodies, bispecific antibodies, and innovative cell therapy approaches including those involving CAR-T-cells. CAR-T-cell therapy, however, might be only suitable for R/R and high-risk FL, due to the high toxicity observed in clinical trials. As ICIs have not, thus far, shown particular promise, there is an urgent need to find new treatment strategies and combinations. In our view, the use of CAR-T-cells (which target FL cells and T_fh_ lymphocytes) and bispecific monoclonal antibodies represents a promising strategy that warrants further research.

## Figures and Tables

**Figure 1 cancers-14-02158-f001:**
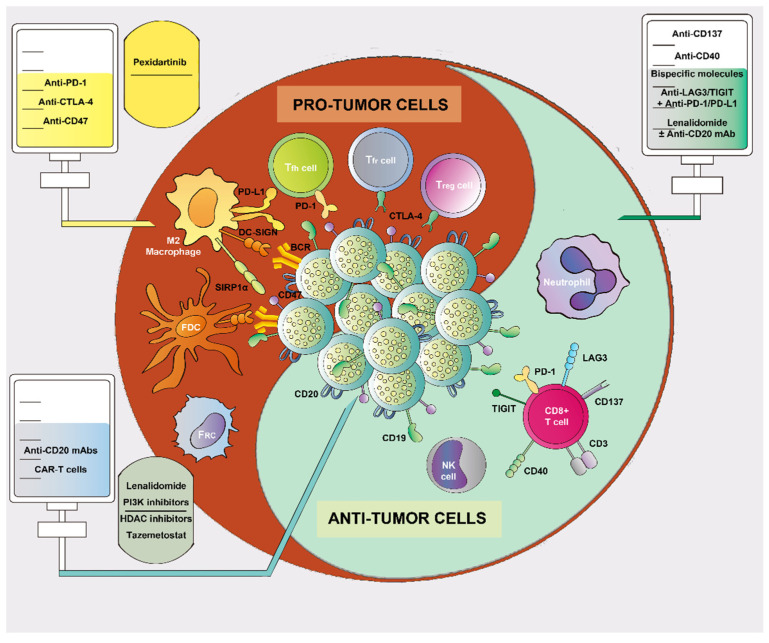
A roadmap of various drugs directed against follicular lymphoma (FL). Therapies against FL cells include anti-CD20 monoclonal antibodies (mAbs), CAR-T-cells, immunomodulatory drugs, PI3K inhibitors, Bruton’s tyrosine-kinase inhibitors, and epigenetic regulators, which can also influence the surrounding microenvironment. Regarding the TME, therapies against cells with pro-tumor activity include immune checkpoint inhibitors, anti-CSF-1, and anti-CD47. These drugs aim to block different targets expressed in T-cells (Treg, Tfh, and Tfr) and macrophages, thus stimulating the immune response. Therapies directed toward TME cells with anti-tumor activity include anti-CD137 and anti-CD40 mAbs (immune checkpoint activators), bispecific molecules, anti-LAG3/TIGIT + anti-PD-1/PD-L1 therapy, and lenalidomide.

**Table 1 cancers-14-02158-t001:** Main cells involved in the tumor microenvironment of follicular lymphoma.

Cells	Phenotype	Secreted Cytokine	Role in FL
T_fr_	CD4+ CD25+ FOXP3+ CXCR5high ICOShigh PD1high BCL6low BLIMP1+	CCL4, IL-16	T_reg_ recruitment. More suppressive than normal T_regs_ Inhibition of CD8+ T-cell activity
T_fh_	CD4+ CD25- CXCR5high ICOShigh BCL6+ PD1high TIM3-	IL-4, IL-17, IL-21, IFNγ	pSTAT6 ↑ T_reg_-recruiting CCL17 and CCL22 production by FL cells FL cell survival and proliferation Inhibition of apoptosis
Stromal cells FDCs CAFs	CD21+, CD23+	CXCL13 CXCL12, CXCL10 (ICAM-1↑), CCL2/19/21, BAFF	Creation of a neoplastic niche Monocyte recruitment and M2-polarization Migration and activation of FL B-cells T-cell recruitment
M2 TAM	CD163+, CD68-	IL-10, IL-15, VEGF	Th2 response Angiogenesis

T_fr_: follicular regulatory T-cell; T_fh_: follicular helper T-cell; FL: follicular lymphoma; FDC: follicular dendritic cell; CAF: cancer-associated fibroblasts; TAM: tumor-associated macrophage.

**Table 2 cancers-14-02158-t002:** Current therapies against FL and TME.

Targeted Drug	Target	Biological Consequences for the TME	Combination	ORR (%)	Status
Pidilizumab	PD-L1	Boost immune synapses	Rituximab	66	Clinical
Nivolumab	PD-1	Boost immune synapses	None	4	Clinical
Ipilimumab	CTLA-4	Boost immune synapses	Rituximab	58	Clinical
Urelumab	CD137	Enhances T-cell and NK antitumor activity	None Rituximab	12 35	Clinical Clinical
Lenalidomide	E3 ubiquitin-ligase cereblon	Immunomodulation, antiangiogenic, antiproliferative effect	Rituximab	74–80	Approved
Ibrutinib	BCR	Crosstalk between tumor cells and macrophages	None Rituximab	37 85	Clinical Clinical
Idelalisib	PI3Kδ	Reduces FDCs-induce genes (angiogenesis, extracellular matrix production). Downregulation of B-T synapses	None Venetoclax	57	Approved Preclinical
Duvelisib	PI3Kγ	M1 polarization	None	42	Approved
Copanlisib	PI3Kαδ	Downregulation of B-T synapses	None	59	Approved
Umbralisib	PI3Kδ, CKIε	Downregulation of B-T synapses	None	45	Approved
Pexidartinib	CSF-1	Diminishes myeloid cell recruitment, M1 polarization	Rituximab	NA	Preclinical
Hu5F9-G4	CD47	Inhibition of phagocytosis	Rituximab	71	Clinical
Tazemetostat	EZH2	Less dependency on FDCs	None	69	Approved
Vorinostat	HDACi	Upregulation of MHCII	None	49	Clinical
Abexinostat	HDACi	Upregulation of MHCII	None	63	Clinical
Mosunetuzumab	CD3/CD20	Cell-mediated cytotoxicity	None	64	Clinical
Glofitamab	CD3/CD20	Cell-mediated cytotoxicity	None	70.5	Clinical
Odronextamab	CD3/CD20	Cell-mediated cytotoxicity	None	92	Clinical
CAR-T-cells	CD19 Anti-CXCR5	Elimination of T_fh_	None None	94 NA	Approved Preclinical

ORR: overall response rate; NA: not available; FDCs: follicular dendritic cells; MHCII: class II majohistocompatibility complex; T_fh_: follicular helper T-cell.

## Data Availability

Not applicable.

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
