# Peer review of "Unveiling the Role of the Tumor Microenvironment in the Treatment of Follicular Lymphoma"

_cancers, 2022, doi:10.3390/cancers14092158_

Round 1

Reviewer 1 Report

Unveiling the Role of the Tumor Microenvironment in the Treatment of Follicular Lymphoma

  1. Blanco, et al

The manuscript discusses the role of the tumor microenvironment on follicular lymphoma covering what is known about the crosstalk between follicular lymphoma cells and cells in the surrounding microenvironment. They specifically cover Treg cells and M2 macrophages (also known as tumor associated macrophages).

Comments to the authors:

  1. On line 39, the authors mention a transformation rate of 2-3% of patients per year. Is this world-wide? In the US? Europe?

  1. On line 86, the authors discuss antigen presentation by B cells and say (on line 88) that centrocytes present antigen to TFH cells  via CD40-CD40L interaction. This is an inaccurate statement since Ag presentation occurs via MHC.

  1. On line 95, the authors mention the importance of CD23 expression on FDCs. CD23 is expressed on B cells. A reference is needed there and a correct statement is also needed.

  1. On line 162 the authors discuss the role of stromal cells and macrophages in FL. This section should be significantly expanded to discuss M1 and M2 macrophages and the fluidity of the 2 populations during different stages of tumor development. A recent review covered this topic (https://www.mdpi.com/1422-0067/22/13/6995).

  1. On line 190, the authors talk about neutrophils and how little is known about their interaction with FL B cells. With only 1 study investigating the role of neutrophils, it should be mentioned that they are predominantly present in peripheral blood.

  1. On line 211 the authors mention genetic alterations in aggressive lymphomas. However FL is indolent with some aggressive disease seen in a smaller population of patients. Please revise.

  1. On line 227, the authors mention pyruvate secretion supporting FL survival. This needs to be expanded to explain more clearly what pyruvate is, which cells secrete it, what levels of it are in FL and how it plays a role.

Author Response

Point 1: On line 39, the authors mention a transformation rate of 2-3% of patients per year. Is this world-wide? In the US? Europe?

Response 1: This transformation rate refers to the US because most recent data comes from there. Maybe this percentage could be extrapolated to the rest of the world, but the latest studies have been performed in the US. There are some old studies performed in Europe (France, for example) that report higher transformation rates.

Point 2:. On line 86, the authors discuss antigen presentation by B cells and say (on line 88) that centrocytes present antigen to TFH cells  via CD40-CD40L interaction. This is an inaccurate statement since Ag presentation occurs via MHC.

Response 2: Ok, I will change it. “On the other hand, when centroblasts stop dividing they become medium-sized cells called centrocytes, and, if the IGV gene mutations have resulted in increased affinity for antigen, they present it to the Tfh cells via the CD40-CD40L interaction class II major histocompatibility complex (MHCII)”.

Point 3. On line 95, the authors mention the importance of CD23 expression on FDCs. CD23 is expressed on B cells. A reference is needed there and a correct statement is also needed.

Response 3: This statement is based on various reviews.

  1. Harris NL. Indolent lymphoma: follicular lymphoma and the microenvironment-insights from the microscope. Hematology Am Soc Hematol Educ Program. 2014 Dec 5;2014(1):158-62. doi: 10.1182/asheducation-2014.1.158. Epub 2014 Nov 18. PMID: 25696849.
  2. Kurshumliu F, Sadiku-Zehri F, Qerimi A, et al. Divergent immunohistochemical expression of CD21 and CD23 by follicular dendritic cells with increasing grade of follicular lymphoma. World J Surg Oncol. 2019 Jul 3;17(1):115. doi: 10.1186/s12957-019-1659-8. PMID: 31269981; PMCID: PMC6610797.
  3. Therapeutic Potential. Int J Mol Sci. 2021 May 19;22(10):5352. doi: 10.3390/ijms22105352. PMID: 34069564; PMCID: PMC8160856.

Point 4. On line 162 the authors discuss the role of stromal cells and macrophages in FL. This section should be significantly expanded to discuss M1 and M2 macrophages and the fluidity of the 2 populations during different stages of tumor development. A recent review covered this topic (https://www.mdpi.com/1422-0067/22/13/6995).

Response 4: Ok, I will explain with more detail the macrophage’s polarization and the characteristics of these cells. I have added new references regarding this topic. 

“TAMs are typically non-phagocytic, with a low proliferation fraction, and they do not undergo apoptosis. They express CD163+, the Fc fragment of IgG and C-type lectin domains. These cells can display two different phenotypes: M1 and M2-polarization. This polarization is controlled by tumor cells within the TME and is dynamic through-out cancer development. M1-macrophages are typically tumor-suppressing cells that act in the TME by recruiting CD8+ T-cells and NK cells. These CD8+ and NK cells express high levels of cytokines and chemokines, recruiting other immune cells and favoring the signaling of anti-tumorigenic pathways (32). On the other hand, the M2-phenotype is associated with tissue remodeling, angiogenesis and progression, making them an at-tractive target for therapies (33). These M2-macrophages secrete immunosuppressive molecules into the TME, suppress T-cell mediated anti-tumor responses and recruit Treg lymphocytes, facilitating tumor proliferation and immune evasion (32,34). TAMs also play a key role in tumor angiogenesis by secreting VEGFA, which can stimulate the chemotaxis of endothelial cells and macrophages (35). In addition, they promote the ep-ithelial–mesenchymal transition process, which enables cancer cells to leave the tissue site, favoring the initiation of metastasis (36). However, TAM’s polarization state is not fixed. At early stages, M1-macrophages are predominant, while the M2-phenotype is more frequent in the advanced stages where tumor proliferation and invasion increas-es(32). It is unknown the role of TAMs in the prognosis of FL. In the pre-rituximab era, some studies suggested that macrophage infiltration was associated with lower survival. Nevertheless, the addition of rituximab to chemotherapy modified their prognostic im-pact, and, although the mechanism is still unknown, it is likely to be associated with an-tibody-dependent cell-mediated toxicity (ADCC)(37,38)”.

Point 5. On line 190, the authors talk about neutrophils and how little is known about their interaction with FL B cells. With only 1 study investigating the role of neutrophils, it should be mentioned that they are predominantly present in peripheral blood.

Response 5: Ok, I will add it to the text. “Little is known about the interactions between FL cells and neutrophils, although they are important players in the innate immune system. Unlike other hematological cells, they are predominantly present in peripheral blood. In preclinical studies, tumor associated neutrophils (TANs) have demonstrated a reduced cytotoxic effect of chemotherapy through the interaction of CD11b/ICAM-1 with CD44 of malignant B-cells”.

Point 6 . On line 211 the authors mention genetic alterations in aggressive lymphomas. However FL is indolent with some aggressive disease seen in a smaller population of patients. Please revise.

Response 6: In the previous paragraph, we explain that there is a subset of patients with FL that evolve with an agressive disease; these genetic alterations detected in gene expresison profiling studies refers to this subset of patients which have an agressive course and may transform into a high-grade lymphoma.  

“The outcome for FL patients dramatically improved with the incorporation of chemo-immunotherapy. However, despite the indolent course of FL, a subset of patients may evolve with rapid disease progression, transforming into high-grade lymphomas. Genetic and individual tumor factors have been associated with the risk of transformation. Regarding individual tumor factors, the intrinsic characteristics of tumor cells, the composition of the TME and patient characteristics are crucial aspects that must be taken into account. Gene expression profiling studies have revealed many genetic alterations that are upregulated in this subset of patients with aggressive diseases and that may be involved in the risk of transformation.

Point 7. On line 227, the authors mention pyruvate secretion supporting FL survival. This needs to be expanded to explain more clearly what pyruvate is, which cells secrete it, what levels of it are in FL and how it plays a role.

Response 7: No problem, I will expand the information on this topic.

Pyruvate is the fundamental metabolite secreted by cancer-associated fibroblasts, resulting in increased use of the citric acid cycle in lymphoma cells and leading to decreased redox regulation. A preclinical study performed in lymph node samples of malignant lymphoma patients, proved that the addition of pyruvate to culture media, with a median concentration of 0.25 mmol/l, contributed significantly to survival of primary FL cells. Further research to uncover the role of CAFs in the lymphoma microenvironment is needed.”

Reviewer 2 Report

Line 64 need to be modified to clarify the most FL are BCL2 positive in contrast with normal GC which are CD1- negative by IHC.

Line 129 needs modification.  Only high grade FL has

Line 485. Should read "BCL2 overexpression' remains....

Author Response

Point 1: Line 64 need to be modified to clarify the most FL are BCL2 positive in contrast with normal GC which are CD1- negative by IHC.

Response 1: ok, I will clarify this concept in the text.

“The histopathology of FL is closely related to key events in normal B-cell development and differentiation. As mentioned before, in approximately 85% of FL cases the malignant B-cells harbor a translocation t(14;18)(q32;q21) that occurs as a consequence of aberrant VDJ recombination in a progenitor B-cell in the BM(18), in contrast with normal GCs which are CD-1 negative by immunohistochemistry”.

Point 2: Line 129 needs modification. Only high grade FL has

Response 2: I can’t find this sentence in the text or maybe I did not understand what yo asked me to change…

Point 3: Line 485. Should read "BCL2 overexpression' remains....

Response 3: ok, no problem. I will change it.  

“While BCL2 overexpression remains the hallmark of FL carcinogenesis, additional aberrations, such as epigenetic mutations, are essential in this context”.